# Study protocol of validating a numerical model to assess the blood flow in the circle of Willis

Yuanyuan Shen ,[1] Yanji Wei,[2] Reinoud P H Bokkers,[3] Maarten Uyttenboogaart,[3,4] J Marc C van Dijk [1]

[1]Department of Neurosurgery, University Medical Center Groningen, Groningen, The Netherlands
[2]Engineering and Technology Institute Groningen, Faculty of Science & Engineering, University of Groningen, Groningen, The Netherlands
[3]Department of Radiology, Medical Imaging Center, University Medical Center Groningen, Groningen, The Netherlands
[4]Department of Neurology, University Medical Center Groningen, Groningen, The Netherlands

**Correspondence to**
Dr Yuanyuan Shen;
y.shen@umcg.nl

## ABSTRACT

**Introduction** We developed a zero-dimensional (0D) model to assess the patient-specific haemodynamics in the circle of Willis (CoW). Similar numerical models for simulating the cerebral blood flow (CBF) had only been validated qualitatively in healthy volunteers by magnetic resonance (MR) angiography and transcranial Doppler (TCD). This study aims to validate whether a numerical model can simulate patient-specific blood flow in the CoW under pathological conditions.

**Methods and analysis** This study is a diagnostic accuracy study. We aim to collect data from a previously performed prospective study that involved patients with aneurysmal subarachnoid haemorrhage (aSAH) receiving both TCD and brain Computerd Tomography angiography (CTA) at the same day. The cerebral flow velocities are calculated by the 0D model, based on the vessel diameters measured on the CTA of each patient. In this study, TCD is considered the gold standard for measuring flow velocity in the CoW. The agreement will be analysed using Pearson correlation coefficients.

**Ethics and dissemination** This study protocol has been approved by the Medical Ethics Review Board of the University Medical Center Groningen: METc2019/103. The results will be submitted to an international scientific journal for peer-reviewed publication.

**Trial registration number** NL8114.

## Strengths and limitations of this study

► This is the first study that intends to validate a lumped haemodynamic model in the CoW in patients with aSAH.
► The to-be-validated model is applicable for clinicians.
► The study can help to understand the cerebral circulation in patients with aSAH.
► The dataset for validation was collected prospectively as a routine clinical procedure.
► The accuracy of transcranial Doppler itself is highly depends on the operator, which limits its accuracy.

perfusion territory[3,4] of major cerebral arties. However, MR demands that subjects are immobile for a longer period and has several contra-indications, such as metal implants. Recently, it was reported that dynamic contrast-enhanced near-infrared spectroscopy could assess CBF at the bedside,[5] which compensates for the disadvantages of MR, but this technique still needs to be validated.

Many fluid dynamic models have been developed to investigate more information about the CBF or to collect the CBF with relatively less demanding on the patient. Cerebral perfusion can be simulated from the level of major cerebral arteries by lumped model[6] to the level of microvascular circulation in the grey and white matter by multiple-network pro-elastic models.[7] In addition, three-dimensional models can simulate haemodynamic parameters, such as wall shear stress and oscillating shear stress,[8–10] as well as that it can predict the haemodynamic situation after a change in arterial network structure.[11,12]

Currently, the application of numerical models is mainly restricted to a laboratory setting, because of its complicated input and simulation, but also due to insufficient validation for use in clinical practice, since many haemodynamic parameters cannot be measured in vivo. The accuracy of simulated

## INTRODUCTION
### Background and rationale

Cerebrovascular disorder (CVD) involves a wide range of vascular disorders that lead to insufficient supply of oxygen and nutrition to the brain. The most severe example is stroke, which has the worst outcome and has become the fifth cause of death in the USA in 2017.[1] Since cerebral blood flow (CBF) is critical during the pathological progress of CVD, the ability to accurately measure CBF is crucial.

Transcranial Doppler (TCD) nowadays has become the most common bedside tool to assess CBF. TCD non-invasively provides real-time CBF velocities in proximal cerebral arteries, including its waveform pattern, but its accuracy is highly operator dependent. MR sequences can provide flow rate[2] and

flow velocity and pressure has been validated with TCD and MR in young volunteers[13–15] and in geriatric suspected stroke patients.[16] Most of the measurements were validated qualitatively. However, in an acute pathological condition, many physiological characteristics of the blood circulation change, for example, viscoelasticity of the vessel wall and hydrostatic pressure. Whether a numerical model still can simulate CBF as accurate as under a normal physiological condition remains unknown.

Therefore, we developed a zero-dimensional (0D) model and aim to use only the circle of Willis (CoW) vessel diameter as input, in order to simulate CBF within the CoW in patients with aSAH. This validation protocol is a fundamental step for subsequent research on the configuration of CoW and CVD.

## Objectives

This study aims to quantitatively validate a 0D model with TCD in patients with aSAH.

## METHODS

We are to collect the data of patients with aSAH who underwent TCD and brain CTA at the same time. Then analysis the agreement between the flow velocity of the CoW simulated by the 0D model and the flow velocity of the CoW measured by TCD. The flowchart of this diagnostic accuracy study is shown in figure 1.

### Study population

For this validation study, we aim to collect data from a previously performed prospective study with 59 adult patients diagnosed with aSAH within 4 days after onset.[17]

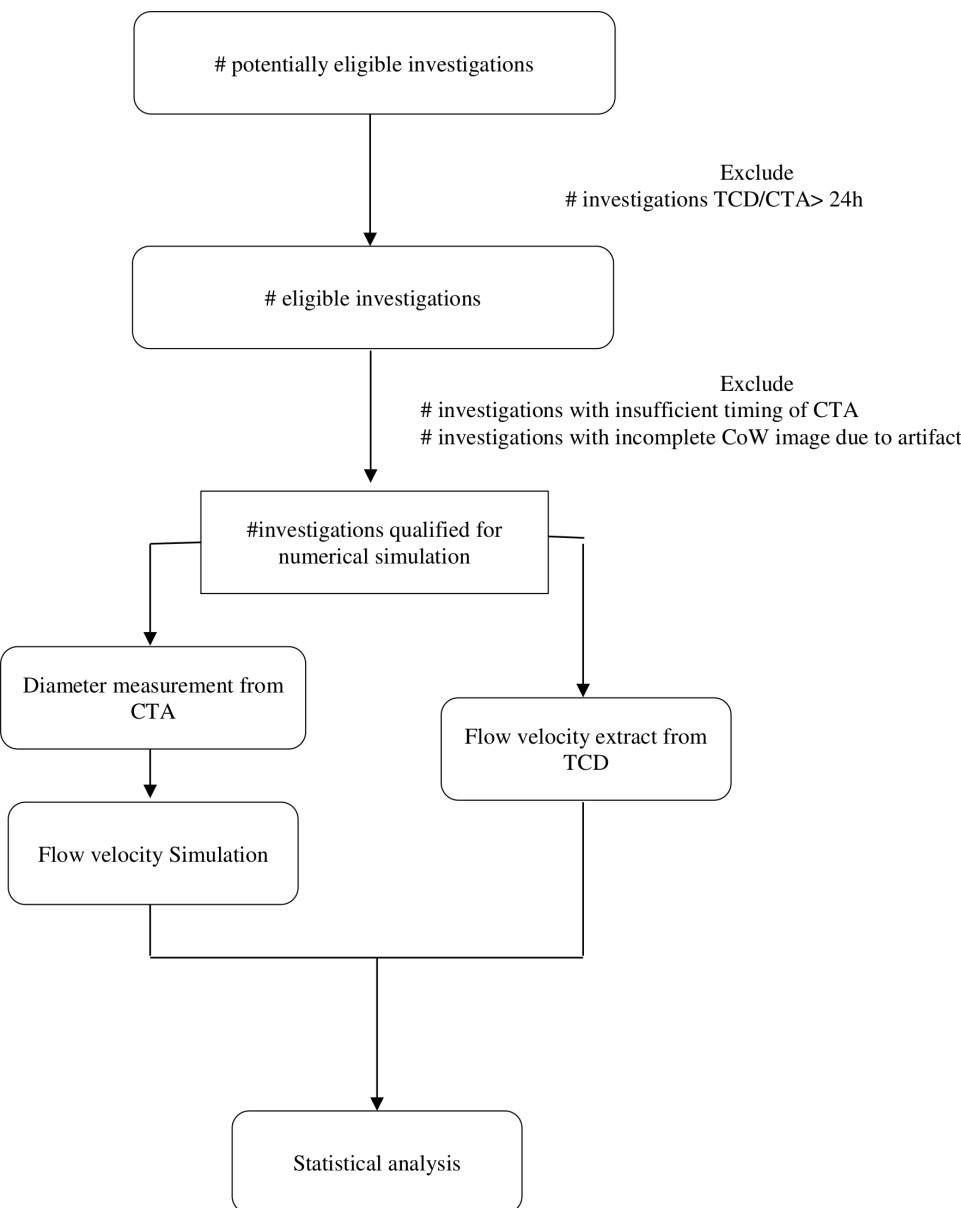

**Figure 1** Study flowchart. CoW, circle of Willis; CTA, CT angiography; TCD, transcranial Doppler.

Moribund patients or patients unable to finish CTA or TCD exam were excluded from this study. Paired TCD and CTA were performed two times during the first 2 weeks after aSAH in every participant.

Paired CTA and TCD performed in one patient at the same time will be considered as one investigation. Investigation is to be excluded if the time interval between CTA and TCD exam is longer than 24 hours. Although patients with aSAH are prone to experience cerebral vasospasm, we assume that the diameter of the arteries of the CoW remains unchanged between CTA and TCD within one investigation.

CTA image quality is to be checked by two board certified radiologists. In case of insufficient CTA image quality, due to coil/clip artefacts or insufficient timing of CTA, investigation will be excluded.

### Flow velocity measurement

TCD was performed by experienced neurophysiology technicians as part of clinical routine. Flow velocities at 11 different locations of 9 arteries of the CoW were reported: (1) distal cervical part of internal carotid artery (ICA); (2) proximal and distal middle cerebral artery (MCA); (3) anterior cerebral artery (ACA); (4) basilar artery (BA); (5) posterior cerebral artery (PCA). Peak systolic velocity; end diastolic velocity and mean velocity were recorded in cm/s.

### Diameter measurement

The diameter of each segment of the CoW is to be measured by semi-automatical central lumen calculation on CTA images, using the workstation TeraRecon AquariusNET iNtuition Viewer, V.4.4.13 .P4. The CTA was performed with iodine contrast agent (Iomeron).

The diameter of each segment of the CoW is to be measured at two points: start and end. The extracranial ICA point is also measured. The start point and end point of each segment are defined as below:

1. Anterior communicating artery (AComA): between left and right ACA. One point in the middle, the other point after the junction.
2. ACA: start after bifurcation of ICA to MCA and ACA, middle at before ACom starts, end after junction of AComA.
3. MCA: start after bifurcation of ICA to MCA and ACA; end before bifurcation.
4. ICA: start at the entrance of the petrous bone (end of C1 part), middle at ophthalmic artery (C6 part); end before bifurcation to MCA and ACA (C7 part).
5. Posterior communication artery (PComA): between ipsilateral ICA and PCA. One point in the middle, the other point after the junction.
6. PCA: start after BA bifurcation to left and right PCA; middle at where it meets PCom, end after the junction.
7. BA: start at the vertebrobasilar junction; end before bifurcation to left and right PCA.

The measuring points for diameters from CTA and for flow velocities from TCD are described in figure 2.

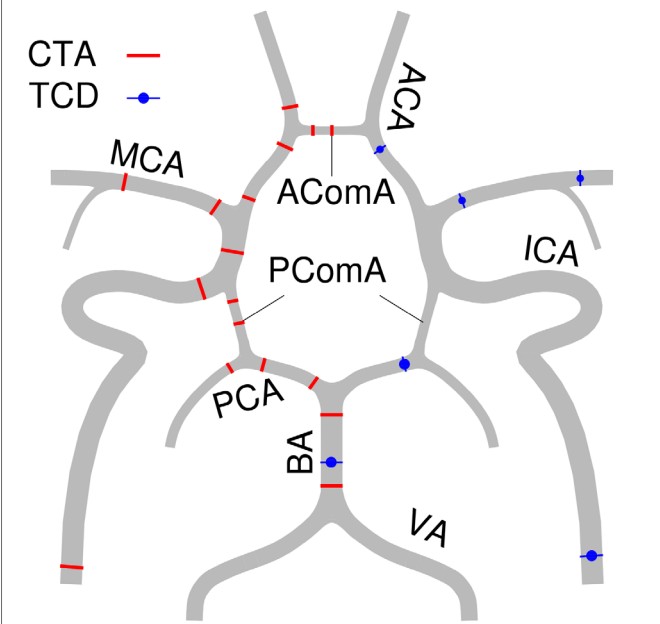

**Figure 2** Schematic diagram of measuring point of diameter from CTA and flow velocity from TCD. CTA, CT angiography; TCD, transcranial Doppler; BA, basilar artery; VA, vertebral artery; PCA, posterior cerebral artery; PComA, posterior communication artery; ICA, internal carotid artery; MCA, middle cerebral artery; ACA, anterior cerebral artery; AComA, anterior communicating artery.

### Flow velocity simulation
#### Hemodynamic model

The haemodynamics in the CoW are described using a lumped parameter model, the 0D model. This model assumes that blood is an incompressible fluid, and that arteries are thin, homogeneous deformable tubes behaving in a viscoelastic manner with a linear Kelvin-Voigt model.[18] The friction losses were calculated with a flow regime (laminar/turbulent)-dependent friction factor using the Haaland approximation. The local losses are neglected in the present model, since accounting such loss in the simulations did not have a significant effect on pressure or flow waveforms.[19]

Different from conventional 0D models that applied the hydraulic–electrical analogue, the present model is based on Simulink, using Simscape Fluids library, which can provide an intuitive and easy way to model the blood circulation system as the hydraulic systems within a block diagram environment (figure 3). This allows us to easily customise the connectivity of the artery network. Each segment in the arterial network is described with 'Segmented Pipeline' block, which accounts its resistive, fluid inertia and wall compliance.

#### Boundary conditions and initialisation

At the root of the ascending aorta, a periodical flow rate that describes the systole with a half sinusoidal wave and the diastole with zero flow rate. At the end of each terminal branch, the present model adopts three-element Windkessel model (WK3) as the downstream condition,

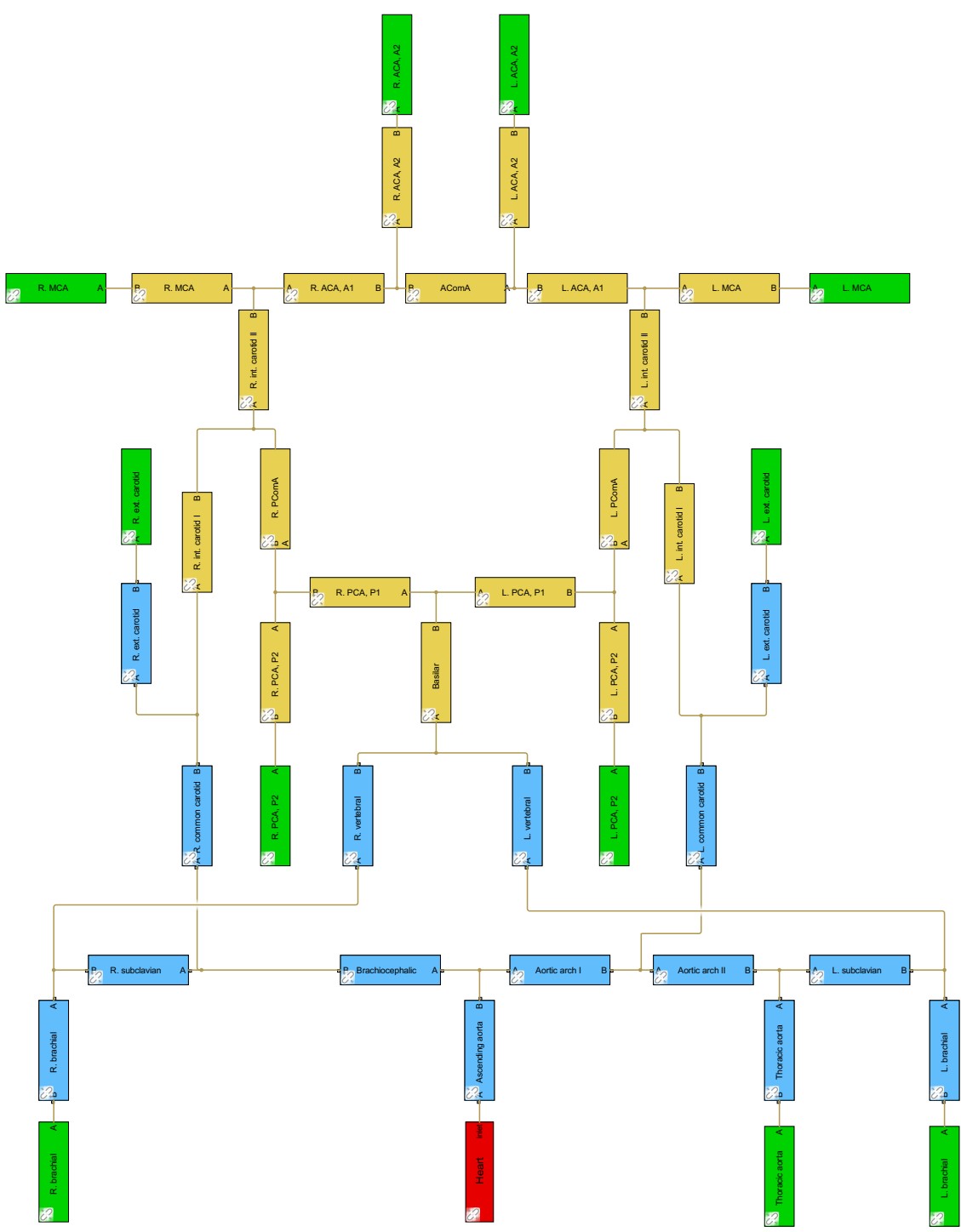

**Figure 3** SBlock diagrams of the arterial network with 33 segments, based on Alastruey *et al.*[13] ACA, anterior cerebral artery; AComA, anterior communicating artery; MCA, middle cerebral artery; PComA, posterior communication artery; PCA, posterior cerebral artery.

which consists of two resistances and a compliance[13] in order to minimise the non-physiological reflection. The WK3 is connected with a constant outlet pressure which represented the pressure at the entrance of the venous system. Sensitivity analysis showed that the results are insensitivity to the outlet pressure, thus a constant pressure of 5 mm Hg is used in the simulation.

The model is initialised with flow rate $Q = 0$ and pressure $P = 0$ in all the segments.

### Physiological data

The basic model in this study consists of an arterial network with 33 segments. The arterial network provides a detailed description of the arterial circulation, particularly the cerebral circulation (figure 2). The geometrical parameters of an artery include length and diameter. The diameter of each segment in the CoW is calculated based on equivalent circular truncated cone volume, using the diameters measured at two ends of the artery from CTA images, while other studies collected diameters based on data from the literature.[15] Since the length of the cerebral arteries ($\sim 10^{-2}$ m) is much shorter than the wavelength of the pulse waves ($\sim 10^0$ m), and measuring length on medical imaging is not straightforward, makes that length is less practical to be used as a diagnostic parameter than diameter. It is, therefore, decided to only use diameters to represent the patient-specific CoW in the simulations in order to the get patient-specific cerebral haemodynamics. The length remains the same as that of Alastruey et al.[13]

The viscoelastic coefficients of the arteries are difficult to be estimated. Ghigo et al[18] investigated the viscoelasticity of the arteries on animals using the Kelvin-Voigt model. They found that the viscoelastic relaxation time (the ratio between the viscoelastic coefficient and the Young's modulus) is nearly constant throughout the network. Due to lacking of in vivo human measurements, the same constant viscoelastic relaxation time as that in Ghigo et al (0.02 s) is to be used in the whole network.

### Data analysis
#### Data collection

For each patient, simulation will run for 10 cardiac cycles to achieve steady-periodic state. The results of the last cycle are to be selected for analysis. The output of the model includes the time series of flow rate and pressure at each segment. The derived mean flow velocity at nine arteries with TCD measurement are collected for statistical analysis.

#### Statistical analysis

All data are to be analysed with Statistical Package of Social Sciences. Lin's Concordance Correlation Coefficient is to be tested first to check the agreement between TCD and the 0D model. If low agreement shows, Pearson's correlation coefficient is to be tested next if the data are normally distributed, the Spearman's rank correlation coefficient is to be tested if data are not normally distributed. Bland-Altman plots are generated to show the agreement between TCD and the simulation by the 0D model.

### Statistical power

There are 118 investigations according to the study design of the previous study before screening. With that sample size, the 95% CI of an expected correlation at 0.8 is 0.73 to 0.86; 95% CI of expected r at 0.6 is 0.47 to 0.70. If 30% investigations are excluded after screening, the 95% CI of expected r at 0.8 is 0.71 to 0.87.

### Patient and public involvement

As mentioned above, the data will be collected from a completed prospective study. The objection register will be checked, and the data from those who objected will be excluded from this study. Thus, no informed consent is needed which has been confirmed by local Medical Ethics Review Board. Patients and public are not involved in the study design.

## ETHICS AND DISSEMINATION

This study protocol has been judged as a study that doesn't fall under the Medical Research Involving Human Subjects Act (WMO) by the Medical Ethics Review Board of the University Medical Center Groningen (METc UMCG) on 12 February 2019 (METc2019/103). Therefore, no patient consent is needed. The previous study "TACTICS" has been approved by the METc UMCG (METc2013/051) and was registered in the Dutch trial register: NTR4157.

**Contributors** YS, as a corresponding researcher, is responsible for the quality of data collection, data analysis and drafting the manuscript. YW developed the numerical model and performed the simulation. MU provided the patients' data, technically assisted the collection of the data and critically revised the manuscript. RPHB critically revised the manuscript. JMCvD supervised the study and critically revised the manuscript.

**Funding** YS receives financial support from China Scholarship Council (CSC, File No. 201706320024).

**Competing interests** None declared.

**Patient and public involvement** Patients and/or the public were not involved in the design, or conduct, or reporting, or dissemination plans of this research.

**Patient consent for publication** Not required.

**Provenance and peer review** Not commissioned; externally peer reviewed.

**ORCID iDs**
Yuanyuan Shen http://orcid.org/0000-0003-0426-5295
J Marc C van Dijk http://orcid.org/0000-0002-0814-5680

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
