## [Reviewer comments · BMJ Open]

ARTICLE DETAILS

TITLE (PROVISIONAL)	Study protocol of validating a numerical model to assess the blood flow in the circle of Willis
AUTHORS	Shen, Yuanyuan; Wei, Yanji; Bokkers, Reinoud; Uyttenboogaart, Maarten; van Dijk, J. Marc C.

VERSION 1 - REVIEW

REVIEWER	Krzysztof Jozwik Institute of Turbomachinery, Faculty of Mechanical Engineering, Lodz University of Technology, Poland
REVIEW RETURNED	09-Jan-2020

GENERAL COMMENTS	There is a possibility to study the flow of blood creating 3D full geometry basing on CT or MRI and simulate the flow. In the paper it has not been comment and the given method has not been proven as the proper one.
---

REVIEWER	Alessandro Stecco Eastern Piedmont University
REVIEW RETURNED	28-Jan-2020

GENERAL COMMENTS	Interesting validation protocol of a numerical model to assess the blood flow iin the circle of willis. Please correct the missing references at page 6 lines 56/46.
---

REVIEWER	Kerem Pekkan Koc University Turkey
REVIEW RETURNED	29-Jan-2020

GENERAL COMMENTS	The objectives need to be made clear. Lumped parameter model returns a large number of data sets. Their analysis and comparison with clinical data requires more thought not fully addressed in the manuscript. Lumped parameter sensitivity study can be performed. This should show the most sensitive cross-sections. Also the outlet pressure 5mmHg venous pressure is not justified. a sensitivity study by varying 5mmHg pressure and documenting their effect would be
---

	useful. Also as you know brain circulation can not be represented by a single WK3 vascular bed as in other organs: Cerebral arteries drain to multiple beds and multiple venous returns emerge. The cerebral circulation is more complex than other organs. I guess you will compare the results but the accuracy of the model results will be greatly undermined due to these simplifications.
--	---

VERSION 1 – AUTHOR RESPONSE

Reviewers' comments to Author:

Reviewer 1

Reviewer Name: Krzysztof Jozwik.

Institution and Country: Institute of Turbomachinery, Faculty of Mechanical Engineering, Lodz University of Technology, Poland.

Comment: There is a possibility to study the flow of blood creating 3D full geometry basing on CT or MRI and simulate the flow. In the paper it has not been comment and the given method has not been proven as the proper one.

Response: We appreciate your review and feedback. The 3D model has been widely used to investigate cerebral flow. The 3D patient-specific simulation can be performed based on topography of medical images, which provides detailed information about blood flow in the CoW, especially in the study of intracranial aneurysm. However, the 3D model is more complicated, time consuming, and generates a lot of data. In terms of the lumped model, it is a rather simple model, that requires less input of patient-specific characteristics (more applicable to daily clinical practice), and it already can offer enough insight for investigating the global characteristic of the blood flow in the CoW. Also, its outcome can be directly compared with TCD measurement. To further clarify this, we have modified our description in the introduction.

Reviewer 2

Reviewer Name: Alessandro Stecco

Institution and Country: Eastern Piedmont University.

Comment: Interesting validation protocol of a numerical model to assess the blood flow in the circle of Willis. Please correct the missing references at page 6 lines 56/46.

Response: Thank you for your careful review of our paper. We corrected the missing reference in the manuscript.

Reviewer 3

Reviewer Name: Kerem Pekkan

Institution and Country: Koc University Turkey

Comment 1. The objectives need to be made clear. Lumped parameter model returns a large number of data sets. Their analysis and comparison with clinical data requires more thought not fully addressed in the manuscript.

Response: Thank you for your comments, which have really helped us during the revision. Based on your feedback we have further clarified the outcome of the lumped model and the processing in data analysis section of the revised manuscript.

Comment 2. Lumped parameter sensitivity study can be performed. This should show the most sensitive crosssections. Also the outlet pressure 5mmHg venous pressure is not justified. a sensitivity study by varying 5mmHg pressure and documenting their effect would be useful.

Response: As stated in diameter measurement section, we measure the diameter of cross-section at two ends of each segment. In patient-specific simulation, the diameter of each segment is calculated based on equivalent circular truncated cone volume, using the two measured diameters. Hence the diameter is not tunable in the model. We have further clarified the diameter calculation in Boundary conditions and initialization section of the revised manuscript. Simulation with various outlet pressure (2-14 mmHg) has been carried out. The following figure showed that the flow rate of each segment in CoW slightly decreases as outlet pressure increases, the difference is less than 3%. It is concussed that the results are insensitive to the outlet pressure.

Figure 1: Flow rate in CoW varying with outlet pressure

We summarized our finding above in the manuscript in Flow rate simulation section.

Comment 3. Also, as you know brain circulation cannot be represented by a single WK3 vascular bed as in other organs: Cerebral arteries drain to multiple beds and multiple venous returns emerge. The cerebral circulation is more complex than other organs. I guess you will compare the results but the accuracy of the model results will be greatly undermined due to these simplifications.

Response: We agree with the reviewer's concern and we have considered this possibility. The cerebral circulation is a very complex system, particularly the auto-regulation under pathological conditions. To address this issue, we plan to run the Lin's Concordance Correlation Coefficient test, as stated in the statistical paragraph of the Method section, to check whether the absolute value from the 0D model and the TCD can be compared directly. If this was not applicable, we will alternately analyze the correlation of their relative value or ratio with Pearson's correlation coefficient/ Spearman's rank correlation coefficient test.